# Factors associated with maternal health services utilization in Pakistan: Evidence from Pakistan maternal mortality survey, 2019

**Farid Midhet, Mubashir Hanif, Samina Naeem Khalid, Raheel Shahab Khan** **\***, **Ikhlaq Ahmad, Shahzad Ali Khan**

Health Services Academy, Islamabad, Pakistan

\* raheel@hsa.edu.pk

## Abstract

### Background

This study investigates the factors associated with maternal health services utilization in Pakistan using two outcome indicators, ideal antenatal care (IANC), defined as the pregnant woman receiving all the essential services included in standard antenatal care, and skilled birth attendance (SBA).

### Methods

This study used the Pakistan Maternal Mortality Survey 2019 data. The study utilized binary logistic regression models to investigate the adjusted association between the outcome variables, separately for IANC and SBA, and the independent variables, education, wealth, parity, and residence.

### Results

Wealth showed a positive association with utilization of IANC (adjusted odds ratio [AOR] = 11.48, 95% CI = 7.76, 16.99) and SBA (AOR = 4.37, 95% CI = 3.30,5. 80). Maternal age was associated only with IANC for women aged 35 or more years (AOR = 1.31, 95% CI = 1.06, 1.62). Increased likelihood of utilization of IANC and SBA services was also observed for women with formal education. Women who had 3–5 previous live births had higher odds of using IANC and SBA than women who had 1–2 or more than five previous live births. Urban residency was not correlated with either IANC or SBA.

### Conclusion

When compared to the wealthy and educated quintile, women in the lower wealth quintile and those without any formal education were less likely to utilize ANC and SBA services. A comprehensive and multipronged approach from the health and education sectors is needed to improve maternal health in Pakistan.

**Data Availability Statement:** All relevant data are within the paper and its Supporting Information files. This study used data that was collected by the

Demographic and Health Surveys Program and is available from www.dhsprogram.com.

**Funding:** The authors received no specific funding for this work.

**Competing interests:** The authors have declared that no competing interests exist.

## Introduction

Pakistan, a lower-middle income country, faces enormous challenges in improving its key maternal health indicators. Public sector spending on health as a percentage of Gross Domestic Product (GDP) is just 1.2%, making out of pocket (OOP) expenditure as the primary mode of healthcare finance [1]. Perinatal mortality rate is 57 deaths per 1,000 pregnancies [2]. Between 2006 and 2019, the maternal mortality ratio (MMR) declined from 276 maternal deaths per 100,000 live births to 186, a mere 33% decline with an average annual rate of reduction of 2.7% [3, 4].

There is a substantial gap between developed and developing countries in the maternal health indicators. According to global estimate of MMR for 2017, about 94% of the global maternal deaths occurred in low- and middle-income countries (LMICs). Also around 68% of the world's total maternal deaths occur in sub-Saharan Africa and another 19% in Southern Asia [5]. Within countries, this skewness of maternal deaths is primarily concentrated in the poor segment. An assessment of maternal mortality between 1990 and 2015 shows that the poorest women accounted for 80% of maternal deaths, up from 68% in 1990 [6]. Poorest women are less likely to access and afford maternal health services than the wealthiest [7, 8].

Maternal health services—which include ANC, SBA, emergency obstetric and neonatal care (EmONC), post-natal care, and family planning—are essential interventions for reducing the risk of maternal and neonatal deaths, and long-lasting morbidity. Ensuring healthy lives of mothers is one of the targets of the Sustainable Development Goals that calls for reducing the global MMR to less than 70 maternal deaths per 100,000 live births by 2030 (SDG 3.1). This requires an average annual rate of reduction of 6.1% (from 2016–2030) globally—an ambitious target for LMICs [5].

The quality and coverage of maternal health services, particularly ANC, are also poor in LMICs. A study on the quality and contents of the ANC among 10 LMICs of Asia and Africa found that although a proportion of pregnant women reported receiving four or more ANC visits, the care provided did not meet the necessary standards for quality [9]. This study also highlighted the importance of developing an ANC indicator that captured the details of quality and content of care. Another study in Rwanda found that uneducated and poor women were less likely to receive high quality ANC, which included all the required components of care [10].

Although the LMICs have made some progress on improving maternal health indicators against their local targets that are specific to their context. In the period 2015–2021, use of SBA was 85%, up from 77% in 2008–2014 and about 20 percentage points from period around 2001 [11]; whereas the use of ANC (at least one visit) by 2016 was 86% [12]. However, the quality of these services lags the coverage and deviates greatly from the recommended checks and guidelines. Despite some kind of high ANC coverage, only half of the women had their blood pressure checked and had their urine and blood tested, according to a study based on the demographic and health surveys (DHS) and multiple indicator cluster surveys (MICs) conducted in LMICs. A third of the women did not receive any of these services at any point during their pregnancy [12]. Studies also show that there are wide variations in the qualifications and competencies of health personnel providing ANC and SBA within and across countries [13].

ANC and SBA are effective interventions for reducing maternal mortality [6]. ANC comprises of the medical procedures during pregnancy to promote and maintain physical and mental health of the mother, preempt and manage complications, and prepare the mother for the birth, postpartum period, and breastfeeding. Focused antenatal care (FANC) is described as an approach that provides for individual assessment and customized decision-making by the provider with an aim for safety and health of the mother and the baby [14].

ANC not only helps in identification of detrimental conditions such as anemia and high blood pressure, it also educates the women on healthy practices, encourages better nutrition during pregnancy, and promotes family planning for birth spacing. ANC also facilitates timely referral to specialized health facility in case of complications during pregnancy and childbirth. Several studies have shown that women using ANC are more likely to use SBA [15–17]. Although ANC is not a sufficient intervention to prevent obstetric complications which are usually unpredictable, it increases the probability of the woman delivering in a health facility where such complications can be identified and addressed.

Pakistan has observed modest increases in SBA and ANC use in recent years. About 69% deliveries were assisted by skilled birth attendants in 2017 (66% occurring in a health facility), up from 52% in 2012; whereas 86% of women received ANC from skilled health providers (doctor, nurse, midwife, or lady health worker) in 2017, up from 73% in 2012 [2, 18]. This paper presents an assessment of pregnant women's access to maternal health services in Pakistan using two indicators, ANC and SBA, and their correlation with the woman's age at birth, wealth, education, parity, and residence. We have used ideal ANC as the care during pregnancy which includes the desired examinations, tests, education and preparedness sessions of ANC. A similar approach has been adapted by other researchers: For example, Arroyave et al. have identified the need to define ANC as a "novel content-qualified ANC coverage indicator" which takes into account the contents and the quality of the care received [19].

## Methods

### Data

We used data from Pakistan Maternal Mortality Survey 2019 (PMMS 2019), publicly available from the DHS program (dhsprogram.com). It is Pakistan's first exclusive national survey on maternal mortality. The survey was designed and conducted by the National Institute of Population Studies (NIPS) (https://nips.org.pk/) in collaboration with the DHS Program. The sample included 136,226 households in the four provinces (Balochistan, Sindh, Khyber Pakhtunkhwa, and Punjab) and the territories of Azad Jammu & Kashmir (AJK), and Gilgit-Baltistan (GB). Births and deaths of last three years were recorded in each household, and 1,177 deaths of women in 15–49 years age-group were investigated using a verbal autopsy questionnaire to identify maternal deaths. Moreover, in a sub-sample of 10% households, 14,703 ever-married women aged 15–49 years were interviewed to identify complications of and health services utilization in pregnancy, delivery, and postpartum in the last three years.

### Description of study variables

Our outcome variables were ideal ANC and SBA. There are concerns about counting a de facto ANC visit as a proper visit and debates on what and how many visits constitute a quality ANC service [20–22]. We defined *ideal* ANC (IANC), opposed to a mere attendance of antenatal clinic, as the ANC that meets all of the following criteria: 1) four or more ANC visits to a skilled healthcare professional during pregnancy, with at least one visit in the first trimester; 2) laboratory tests of blood and urine carried out at least once; 3) blood pressure measure on each visit; 4) prescription of iron tablets; 5) counselling on nutrition during pregnancy; and 6) at least one dose of tetanus toxoid vaccination. SBA was defined as the delivery conducted by a skilled birth attendant including doctor, nurse, midwife, community midwife, or Lady Health Visitor (LHV).

We identified woman's age at birth ($< 25$ years, 25–34 years, $\geq 35$ years), education (no schooling, less than $10^{th}$ grade, $10^{th}$ grade or higher), wealth quintiles, parity (1–2, 3–5, and $\geq 5$ previous live births), and urban/rural residence as independent variables to separately

estimate their effects on each of the two outcome variables. Both types of variables were created through re-coding of the raw data available from the PMMS 2019 data files.

## Statistical analysis

We generated cross-tabulation between the dependent and independent variables, estimating the significance of the relationship through unadjusted odds ratio and Chi-squared test. We used binary logistic regression models to investigate the adjusted association between the outcome variables (separately for IANC and SBA) and the independent variables (education, wealth, and residence), after adjusting for the effects of age, parity, and prior history of poor pregnancy outcomes. The binary logistic regression results generated adjusted odds ratios (exponentiation of beta coefficients) and their 95% confidence intervals. Diagnostic tests on binary logistic regression analyses were also carried out including the Hosmer-Lemeshow Goodness of Fit Test. For testing collinearity between independent variables, we used the raw data (where age, parity, education, and wealth index were continuous variables) and conducted linear regression analysis keeping parity as dependent variable and calculating the correlation matrix and variance inflation factor. All these tests were found to be within acceptable limits. The statistical significance (p-value) is also reported. We used SPSS (v.19) for estimating the results.

## Results

Table 1 shows the use of IANC and SBA services by sociodemographic factors. The overall proportion of women who received IANC was only 20.5%. The descriptive results indicated that the proportion receiving IANC and SBA services was higher among women who were in the age group of 25–34 years, had 10th grade or higher education, were in the highest wealth quintile and had 1–2 children.

Table 2 presents the regression results for association between IANC and SBA use and sociodemographic factors. Wealth quintile showed a significantly positive association with IANC and SBA use. The odds of using IANC services were 11.48 times [95% CI = 7.76, 16.99] more for women in the highest wealth quintile than for women in the lowest quintile. The odds of SBA use also increased with each increase in the wealth level.

Age at birth was significantly associated only with IANC use for women aged 35 or more years [adjusted odds ratio = 1.31, 95% CI = 1.06, 1.62]. It, however, was not associated with SBA. Increase in odds ratio for utilization of IANC and SBA services were also observed for women with education. Women who had 3–5 child births had higher odds of using IANC and SBA than women had 1–2 or more than five births. Women who had 1–2 child births were less likely to use IANC and SBA. Place of residence didn't show a significant association with IANC and SBA.

## Discussion

The existing evidence on the effect of sociodemographic factors on utilization of maternal health services in Pakistan is based on a *de facto* definition of ANC, which counts a visit during pregnancy to a healthcare provider as ANC regardless of the contents and quality of the visit. We used a comprehensive definition of ANC that includes the essential medical check-ups, laboratory tests, and communication with healthcare provider to reflect the completeness and quality of ANC. We also used a nationally representative sample from the first-ever exclusive survey on maternal mortality and morbidity in Pakistan.

The purpose of this study was to assess the socio-demographic factors affecting maternal healthcare utilization represented by ANC and SBA. Several studies have explored the factors

**Table 1. Descriptive statistics on use of maternal health services, %.**

|  | IANC | SBA |
|---|---|---|
| **Age at birth** |  |  |
| < 25 years | 18.9 | 70.9 |
| 25–34 years | 22.5 | 70.7 |
| ≥ 35 years | 17.1 | 64.3 |
| **Education** |  |  |
| No schooling | 7.5 | 58 |
| < 10th grade | 22.6 | 76.6 |
| 10th grade or higher | 43.2 | 84.6 |
| **Wealth quintile** |  |  |
| Lowest | 3 | 45.9 |
| Q2 | 10 | 62 |
| Middle | 17.7 | 74 |
| Q4 | 31.2 | 82.9 |
| Highest | 47.2 | 88.6 |
| **Parity** |  |  |
| 1–2 births | 27.3 | 76.2 |
| 3–5 births | 19 | 69.1 |
| ≥ 6 births | 6.5 | 53.2 |
| **Residence** |  |  |
| Urban | 27.9 | 78.1 |
| Rural | 15.1 | 63.2 |
| **Region of residence** |  |  |
| AJK | 35.8 | 77.2 |
| GB | 18.9 | 61.3 |
| Balochistan | 3.5 | 53.1 |
| KP | 17.2 | 70.3 |
| Sindh | 16.8 | 67.3 |
| Punjab | 27.9 | 78.3 |

IANC, ideal antenatal care

SBA, skilled birth attendance

Urban women were nearly two times higher, 28%, to receive IANC than rural women (15%). It was found that women with 10th grade or higher education were about one and a half times more likely (85%) to receive assistance from SBA during birth than the uneducated mothers (58%).

affecting ANC visits, but there exists, to our knowledge, no study in Pakistan that uses *ideal* ANC. There is also little empirical evidence around the factors affecting SBA in Pakistan. We have shown that ANC utilization and use of SBA have strong association with mother's wealth, education, and parity.

According to the Pakistan Demographic and Health Surveys (PDHS), ANC visits have increased over the years. The proportion of women using ANC (for the most recent birth) was 61% in 2006–07, 73% in 2012–13, and 86% in 2017–18 [2]. However, only 51% of the women had four or more visits in 2017–18 as recommended by the WHO's 2002 focused ANC (FANC) model [2, 23]. The components of ANC, as reported in PDHS 2017–18, were not uniform across the visits for all women. Only 55% women had ANC visit in the first trimester, 89% percent had their blood pressure checked, 70% had their urine and blood samples taken, 69% received tetanus toxoid vaccination, and 70% received counselling on balance diet [2].

**Table 2. Associations of sociodemographic outcomes with IANC and SBA utilization.**

| | IANC | | SBA | |
| --- | --- | --- | --- | --- |
| | AOR | conf. interval | AOR | conf. interval |
| **Age at birth** | | | | |
| < 25 years (ref.) | - | - | - | - |
| 25–34 years | 0.80 | 0.67–0.96 | 0.98 | 0.83–1.16 |
| ≥ 35 years | 1.31** | 1.06–1.62 | 1.14 | 0.96–1.36 |
| **Education** | | | | |
| No schooling (ref.) | - | - | - | - |
| 1–9 grades | 1.88** | 1.53–2.31 | 1.32* | 1.13–1.55 |
| ≥ 10th grade | 3.24** | 2.63–3.99 | 1.28* | 1.04–1.56 |
| **Wealth quintile** | | | | |
| Lowest (ref.) | - | - | - | - |
| Second | 2.75** | 1.88–4.02 | 1.67** | 1.41–1.97 |
| Middle | 4.18** | 2.87–6.08 | 2.52** | 2.08–3.05 |
| Fourth | 7.17** | 4.91–10.48 | 3.45** | 2.74–4.33 |
| Highest | 11.48** | 7.76–16.99 | 4.37** | 3.30–5. 80 |
| **Parity** | | | | |
| 1–2 births (ref.) | - | - | - | - |
| 3–5 births | 2.71** | 1.94–3.79 | 1.72** | 1.38–2.14 |
| ≥ 6 births | 1.81** | 1.32–2.47 | 1.33** | 1.11–1.60 |
| **Residence** | | | | |
| Rural (ref.) | - | - | - | - |
| Urban | 0.95 | 0.82–1.11 | 1.10 | 0.95–1.26 |

* $p < 0.05$

** $p < 0.001$

AOR, adjusted odds ratio

Adjusted variables include: mother's age at birth, wealth, education, parity, and place of residence

IANC, ideal antenatal care

SBA, skilled birth attendance

Considering these elements of the optimal ANC, the PMMS 2019 data showed that just 20.5% of women obtained adequate ANC (defined as IANC in this study).

It was observed that wealth was the strongest predictor of ANC and SBA use. This highlights the substantial out-of-pocket expenses in Pakistan, which pose financial barriers for poor women seeking quality maternal health services [24]. Several studies within Pakistan and other developing countries support positive association between wealth and 'ANC and SBA' use [25–29]. This suggests inequity in service utilization imposed by wealth status and that removing the financial barriers to the poor can increase uptake of ANC and SBA. As Pakistan gains significant improvement in the coverage these important indicators of ANC and SBA, the questions about equity and quality of care still remain. This could be one reason that Pakistan has not seen a corresponding decrease in maternal, perinatal, and neonatal mortality–having the highest levels of these indicators among all its neighbouring countries except Afghanistan. There is a need to reach out to the poor and the rural women who do not have access to modern medical care during pregnancy and childbirth. Moreover, we found that only about one fifth of the pregnant women received ANC that was appropriate and complete, and about 80% of the pregnant women who seek ANC are unable to receive the full benefits of

ANC because of the poor quality of care in the health facilities. This should be a cause of concern for the policymakers.

Studies show that demand side financing (DSF) can improve utilization of maternal health services. For instance, the maternal health voucher programmes in Bangladesh and Kenya were associated with increase in facility-based deliveries [30, 31]. Similarly, a study on the effect of a voucher scheme in a rural district of Pakistan showed a positive association with facility-based deliveries [32]. At present, Pakistan has started demand-side healthcare financing programs on federal and provincial levels: Sehat Sahulat Program (federal, Punjab, Azad Jammu & Kashmir, Gilgit Baltistan), Sehat Card Plus (Khyber Pakhtunkhwa). These are insurance programs—financed with general tax revenues—that provide free healthcare services, including facility-based delivery services to the beneficiaries [33, 34]. Access to maternity services—caesarean and normal delivery—is universal in Khyber Pakhtunkhwa, whereas the federal program is also planning to make it universal in Punjab and areas administered by the federal government. As the programs have enrolled all the poorest population in the respective regions, it could be expected that the proportion of women using institutional deliveries will increase among women in these poor wealth quintiles. However, high risks areas in Sindh and Balochistan do not have such programs in place and the proportion of women receiving IANC and SBA is also the lowest in these provinces. The findings suggest that reducing the financial barriers to maternity services for women could improve uptake of ideal ANC and coverage of SBA.

Woman's age at birth may affect the utilization of maternal health services; however, the results may be context based. This study observed that the odds of using IANC by women of age 35 or more years were higher compared to women aged less than 25 years. This may be due to the fact that women over 35 have higher risks, which prompts them to seek ANC and be cautious about their health; moreover, because healthcare practitioners pay more attention to these women, a higher percentage of them receive optimal ANC. However, the contents and quality of ANC must not be driven only by the risk profile of pregnant women. Similarly, our results showed that women with more than two childbirths were more likely to use ANC and SBA services as compared to women with their first-born or second-born child. This finding is in contrast with the general view by several studies that ANC and SBA utilization is higher for women with first-born child (primipara women) than multiparous women [29, 32, 35, 36]. According to our results the odds of using ANC and SBA are higher among women having three to five births than women having one or two births. One plausible explanation for this is that women who have faced complications or unpleasant experience in their first birth are more likely to use ANC and SBA, and also the attention of healthcare providers in the form of ideal ANC [37].

The study's results showed a positive association between women education and utilization of ANC and SBA services, which is in line with numerous other studies that highlight the positive effect of women education on the utilization of maternal health services [16, 17, 25, 26, 37]. Generally, education broadens the understanding of women about the health risks, warnings, and healthy practices during pregnancy. It also gives women more autonomy to make health related decisions for themselves [38]. Such autonomy may be reinforced by the financial status of women, as educated women are likely to be in well-off families and/or work in a formal sector compared to uneducated women. The positive role of education highlights the importance of provision of school-based education to adolescent girls. According to the PMMS data, 52% of the interviewed women had no education and the proportion of women with no education in the lowest quintile was a staggering 91% [4]. There is a need to invest in girl's education for long-term health outcomes and close the information gap about maternal health among uneducated adult women through local media (radio, television), health promotion activities, and maternal and child health (MCH) programs.

We tested the presence of multicollinearity between the independent variables, particularly wealth and education and parity and woman's age at birth, which were within acceptable limits. Nonetheless, our results should be interpreted with caution. It may be noted, however, that even though these variables are commonly interrelated, their independent effects on pregnancy outcomes have also been established.

This study has certain limitations: Since this was a household survey, the exact parameters of an ideal ANC could not be captured. We did not use the WHO ANC model of eight ANC visits with a SBA, (recommended to lower perinatal mortality and enhance the woman's experience with healthcare), as it would further limit the number of women receiving ideal ANC in our setup. Hence the ideal ANC definition adopted in this study may be regarded as the bare minimum that a pregnant woman should receive. We did not control for a poor pregnancy outcome in the past (pregnancy loss and neonatal death), which may play a role in increasing the uptake of maternal health services to avoid unpleasant experiences in future. We did not include other factors contributing to the socioeconomic status of the household, such as husband's education and occupation, because we used the wealth index and quintiles reported by the survey, which reflects the overall socioeconomic status of the household. Finally, we have included in our model both the wealth quintiles and woman's education as independent variables determining the likelihood of ANC and SBA use. We believe that the woman's education is an important determinant of her empowerment and ability to use health services independent of the household's wealth quintile.

## Conclusions

The most important message from this study is that, only one-fifth of pregnant women in Pakistan receive comprehensive ANC, and that poorer women and those with low education are less likely to receive proper antenatal care and skilled birth attendance. It seems that, despite the significant increase in the proportion of pregnant women visiting the health facilities for ANC, the care provided to them is incomplete and of poor quality. The federal and provincial governments need to improve the quality of ANC through health systems strengthening, training and supervision of healthcare providers, and by expanding the network of health insurance for the poor. In this regard, Sehat Sahulat Program can be utilized which provides improved access and good quality medical services to the poor population through a micro health insurance scheme.

There is also a need to develop and enforce guidelines for healthcare providers, both in the public and private sectors, to encourage them to provide complete and high-quality ANC services to the women. Lady Health Workers and other community-based workers should educate pregnant women about the importance of a comprehensive ANC including lab tests, counselling, tetanus immunization, and iron supplementation. Our study only provides initial insights into this important and persistent problem of reproductive health services in Pakistan and further work should be done for this important cause. Along with quality of care, improved SBA coverage and more women reaching SBAs will lead to better outcomes and improved IANC services.

## Supporting information

**S1 Dataset.**
(SAV)

**S2 Dataset.**
(SAV)

## Acknowledgments

We extend our thanks to National Institute of Population Studies (NIPS), ICF and the DHS Program for collecting and providing the data that made this study possible.

## Author Contributions

**Conceptualization:** Farid Midhet, Mubashir Hanif.

**Data curation:** Farid Midhet.

**Formal analysis:** Farid Midhet.

**Investigation:** Farid Midhet.

**Methodology:** Farid Midhet.

**Writing – original draft:** Mubashir Hanif, Raheel Shahab Khan.

**Writing – review & editing:** Farid Midhet, Samina Naeem Khalid, Ikhlaq Ahmad, Shahzad Ali Khan.

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
