## [Decision Letter · Decision Letter 0]

10 Jan 2023

PONE-D-22-31027Factors associated with maternal health services utilization in Pakistan: evidence from Pakistan Maternal Mortality Survey, 2019PLOS ONE

Dear Dr. Khan,

Thank you for submitting your manuscript to PLOS ONE. After careful consideration, we feel that it has merit but does not fully meet PLOS ONE’s publication criteria as it currently stands. Therefore, we invite you to submit a revised version of the manuscript that addresses the points raised during the review process.

Kindly refer to the peer review findings shared at the end of the email.

We look forward to receiving your revised manuscript.

Kind regards,

**Dr. Syed Khurram Azmat**, PhD, MPH, MD

Academic Editor

PLOS ONE

Journal Requirements:

When submitting your revision, we need you to address these additional requirements. 1. Please ensure that your manuscript meets PLOS ONE's style requirements, including those for file naming. The PLOS ONE style templates can be found at https://journals.plos.org/plosone/s/file?id=wjVg/PLOSOne_formatting_sample_main_body.pdf and https://journals.plos.org/plosone/s/file?id=ba62/PLOSOne_formatting_sample_title_authors_affiliations.pdf****

Reviewers' comments:

Reviewer's Responses to Questions

**Comments to the Author**

1. Is the manuscript technically sound, and do the data support the conclusions?

Reviewer #1: Yes

Reviewer #2: Partly

2. Has the statistical analysis been performed appropriately and rigorously? 

Reviewer #1: Yes

Reviewer #2: Yes

3. Have the authors made all data underlying the findings in their manuscript fully available?

Reviewer #1: Yes

Reviewer #2: No

4. Is the manuscript presented in an intelligible fashion and written in standard English?

Reviewer #1: Yes

Reviewer #2: No

5. Review Comments to the Author

Reviewer #1: This is an important contribution and should be published. However, there are suggestions for the authors that would (in my opinion) make the article more useful and interesting

Overall Comments:

Key points of uptake of IANC and SBA and their determinants are well made

The overall article can be condensed into a shorter version that is more pithy but also more readable.

Technical Comments:

Given that IANC is an important concept and likely one that the authors have constructed (or used a novel approach), it would be good to describe a bit better how it was determined what is IANC. For eg, the diff between one or more of the 6 criteria, did they apply some score and a cut off etc. or was it if so and so criteria were met

May be useful to think of a comparison between de facto def of ANC vs. IANC (suggestion, not necessarily for this paper)

Would really like an exploration of colinearity between determinants such as education and wealth. Test for colinearity may be reported as part of results section

Discussion may be revisited. While there is some discussion of the determinants and their international correlation, it would serve to go a bit more in depth about their implications.

Also, authors may wish to comment on the intersection of these determinants.

Plus a discussion between discordance of the determinants of IANC and SBA may be interesting

As an editorial point, would suggest moving suggestions towards the end. Plus some suggestions such as those related to the Sehat Sahulat Program (national health insurance) may be taken in the context of current limitations - funding plus the fact that it is only for inpatients

Reviewer #2: Overall the paper is structured well. However, what one expects from a secondary analysis paper is the inference from results and a set of recommendations with a way forward.

Neither discussion nor conclusion give any solid recommendations and a way forward.

Limitations are part of Discussion, not the Conclusion.

Conclusion section again discusses the results which is unnecessary. Only last line gives a very generic recommendation which should be unpacked to inform the policy makers and program managers: what is meant by multisectoral approach from health & education sector? Is there any example from Pakistan or region from where a model may be picked up and replicated /adapted.

This is a very important topic so authors must give thread bear details.

6. PLOS authors have the option to publish the peer review history of their article (what does this mean?). If published, this will include your full peer review and any attached files.

Reviewer #1: No

Reviewer #2: No

---

## [Author Response · Author response to Decision Letter 0]

17 Mar 2023

Dear Editor,

I am grateful for the reviewers' suggestions. In the revised manuscript, we have tried to address the remarks as best as possible. We have responded to each remark in the 'Response to Reviewer' document.

---

## [Decision Letter · Decision Letter 1]

29 May 2023

PONE-D-22-31027R1Factors associated with maternal health services utilization in Pakistan: evidence from Pakistan Maternal Mortality Survey, 2019PLOS ONE

Dear Dr. Khan,

Thank you for submitting your manuscript to PLOS ONE. After careful consideration, we feel that it has merit but does not fully meet PLOS ONE’s publication criteria as it currently stands. Although, the authors have made significant changes and improved the manuscript accordingly. However, there are still some minor changes which requires attention before the journal can take the final decision. Therefore, we invite you to submit a revised version of the manuscript that addresses the points raised during the review process

We look forward to receiving your revised manuscript.

Kind regards,

**Syed Khurram Azmat**, PhD, MPH, MD

Academic Editor

PLOS ONE

Journal Requirements:

Reviewers' comments:

Reviewer's Responses to Questions

**Comments to the Author**

1. If the authors have adequately addressed your comments raised in a previous round of review and you feel that this manuscript is now acceptable for publication, you may indicate that here to bypass the “Comments to the Author” section, enter your conflict of interest statement in the “Confidential to Editor” section, and submit your "Accept" recommendation.

Reviewer #1: (No Response)

Reviewer #2: All comments have been addressed

2. Is the manuscript technically sound, and do the data support the conclusions?

Reviewer #1: Partly

Reviewer #2: Yes

3. Has the statistical analysis been performed appropriately and rigorously? 

Reviewer #1: Yes

Reviewer #2: Yes

4. Have the authors made all data underlying the findings in their manuscript fully available?

Reviewer #1: Yes

Reviewer #2: Yes

5. Is the manuscript presented in an intelligible fashion and written in standard English?

Reviewer #1: Yes

Reviewer #2: Yes

6. Review Comments to the Author

Reviewer #1: The authors have improved upon the original submission by addressing the discussion section. However, some important considerations remain and must be addressed:

1. They show that only around a fifth of the wowmen avail IANC while 2/3 avail SBA. Statistically but more importantly contextually that makes these two very different phennomena. For example, age seems not to change SBA and there is no dose-dependance of education or parity suggesting (when there is a dramatic dose effect of over 6 children elsewhere), demand issues. It would useful for the authors to describe what is similar and different, in terms of which women avail which service, location, demographic and the channel of service delivery

2. A recommendation is made for reaching poor and rural women. However, their results show no difference in utilization by location. Further, given that over half of Pakistan's urban population lives in an urban slum (which would not be identified in this survey, it is likely that their being underserved is not being highlight (given the wealth and educaiton associations that are described). This would suggest that the recommendation should be to focus predominantly on urban poor

3. There is some discussion of supply side interventions. It is unclear given the correlates for low IANC that are described in the regression would mean much more utilization of better trained or equipped providers/ facilities would lead to higher uptake. It seems from the results presented that the problem is a lack of demand and recommendations may address that.

4. A similar argument may be made for SBA. Pakistan saw rapid rise in uptake of SBA between 2012 and 2017, that was attributed by many to supply side interventions such as better equipped and 24/7 facilities. That advantage has been realized. Perhaps a more nuanced discussion of demand creation for both IANC and SBA, given demographic features of low utilizeres may be proposed

5. The disussion is still a bit long, repetitive (both of results and arguments) and circular (argument order need to be revisited for lovgical flow). It can be cut by at least 1/3

6. I am not sure I agree with supply side recommendations that are presented, given the above points. These must be revisited to include a more sophisticated inclusion of context and demand issues

Reviewer #2: Authors have made revisions that are acceptable and changes made are fine for the manuscript to be accepted.

7. PLOS authors have the option to publish the peer review history of their article (what does this mean?). If published, this will include your full peer review and any attached files.

Reviewer #1: No

Reviewer #2: **Yes: **Babar Tasneem Shaikh

---

## [Author Response · Author response to Decision Letter 1]

5 Jul 2023

REVIEW COMMENTS TO THE AUTHOR

Reviewer #1: The authors have improved upon the original submission by addressing the discussion section. However, some important considerations remain and must be addressed:

Reviewer #1: Comment

1. They show that only around a fifth of the women avail IANC while 2/3 avail SBA. Statistically but more importantly contextually that makes these two very different phenomena. For example, age seems not to change SBA and there is no dose-dependance of education or parity suggesting (when there is a dramatic dose effect of over 6 children elsewhere), demand issues. It would useful for the authors to describe what is similar and different, in terms of which women avail which service, location, demographic and the channel of service delivery.

Answer: 1

There are 2/3 of women who have availed SBA services, but it does not mean that they had an Ideal ANC (IANC). We have explained that the trends of increasing use of ANC and SBA are parallel. However, the quality of ANC received by women is in question, which we wish to highlight. Ideal ANC means that they have the ANC visits & the required investigation in addition to the utilization of SBA services, the standards as explained below: 

• four or more ANC visits to a skilled healthcare professional during pregnancy, with at least one visit in the first trimester; 

• laboratory tests of blood and urine carried out at least once; 

• blood pressure measure on each visit; 

• prescription of iron tablets; 

• counselling on nutrition during pregnancy; and 

• at least one dose of tetanus toxoid vaccination. SBA was defined as the delivery conducted by a skilled birth attendant including doctor, nurse, midwife, community midwife, or Lady Health Visitor (LHV).

 We argue that the quality of ANC is more important than simply visiting the health facility during pregnancy without being properly checked for potential risk factors, receiving iron-folic acid supplementation, getting anti-tetanus vaccination, and being informed about nutrition.

It would useful for the authors to describe what is similar and different, in terms of which women avail which service, location, demographic and the channel of service delivery.

The women who use SBA services are mostly affected by wealth quintile and education. The discussion points is that women in Baluchistan, Sindh and GB have relatively lower SBA services because of the longer distances to health facilities although other factors like parity and education are also in play.

Reviewer #1: Comment

2. A recommendation is made for reaching poor and rural women. However, their results show no difference in utilization by location. Further, given that over half of Pakistan's urban population lives in an urban slum (which would not be identified in this survey, it is likely that their being underserved is not being highlight (given the wealth and education associations that are described). This would suggest that the recommendation should be to focus predominantly on urban poor

Answer: 2

We have pointed out that both poverty and residence in rural areas are independent risk factors, and should be addressed. This impact is independent of the province of residence. There is definitely marked difference in rural and urban women. 

a) The table 1 shows that urban women utilize IANC (27.9%) and rural women use (15.5) which shows significant difference.

b) Our study suggests strong relationship with lowest wealth quintile (only 3% IANC)

c) Yes, definitely urban poor should be the focus in a province, but since the data of PMMS was not segregated in the urban slums therefore this can be a limitation of the present study.

Reviewer #1: Comment

3. There is some discussion of supply side interventions. It is unclear given the correlates for low IANC that are described in the regression would mean much more utilization of better trained or equipped providers/ facilities would lead to higher uptake. It seems from the results presented that the problem is a lack of demand and recommendations may address that.

Answer: 3

We respectfully disagree and say that this is not the impression coming out of our results. We are focusing on quality of ANC rather than simply the supply of it; hence, it is a supply-side issue. If we will stress on IANC services, that will automatically include SBA services.

Additionally, if the pregnant women reach the health facilities and interact with the health system including a contact with the SBAs, this would increase their likelihood of opting for institutional delivery rather than home delivery by an untrained birth attendant. 

Reviewer #1: Comment

4. A similar argument may be made for SBA. Pakistan saw rapid rise in uptake of SBA between 2012 and 2017, that was attributed by many to supply side interventions such as better equipped and 24/7 facilities. That advantage has been realized. Perhaps a more pronounced discussion of demand creation for both IANC and SBA, given demographic features of low utilizers may be proposed.

Answer: 4

Our focus is quality of ANC. However, we have mentioned interventions like Sehat Sahulat Program in the discussion and conclusion.

Reviewer #1: Comment

5. The discussion is still a bit long, repetitive (both of results and arguments) and circular (argument order need to be revisited for logical flow). It can be cut by at least

Answer: 5

We emphasize the importance of giving attention to the association between each factor and ANC.

Reviewer #1: Comment

6. I am not sure I agree with supply side recommendations that are presented, given the above points. These must be revisited to include a more sophisticated inclusion of context and demand issues

Answer: 6

Definitely, the supply side recommendation is needed, which we have highlighted that through LHWs and other community-based workers efforts should be made to improve knowledge of women about the importance of ANC services. There should also stress on the importance of blood tests and other necessary services. 

Reviewer #2: Authors have made revisions that are acceptable and changes made are fine for the manuscript to be accepted.

---

## [Decision Letter · Decision Letter 2]

6 Sep 2023

PONE-D-22-31027R2Factors associated with maternal health services utilization in Pakistan: evidence from Pakistan Maternal Mortality Survey, 2019PLOS ONE

Dear Dr. Khan,

Thank you for submitting your manuscript to PLOS ONE. After careful consideration, we feel that it has merit but does not fully meet PLOS ONE’s publication criteria as it currently stands. Therefore, we invite you to submit a revised version of the manuscript that addresses the points raised during the review process.

In addition to the feedback from two peer reviewers, the authors are recommended to acquire content editing services from a native English language editor for copy editing, proofing, grammatical, spelling, and other common language errors in order to improve overall language of the manuscript.

We look forward to receiving your revised manuscript.

Kind regards,

**Dr. Syed Khurram Azmat**, PhD, MPH, MD

Academic Editor

PLOS ONE

Journal Requirements:

Reviewers' comments:

Reviewer's Responses to Questions

**Comments to the Author**

1. If the authors have adequately addressed your comments raised in a previous round of review and you feel that this manuscript is now acceptable for publication, you may indicate that here to bypass the “Comments to the Author” section, enter your conflict of interest statement in the “Confidential to Editor” section, and submit your "Accept" recommendation.

Reviewer #3: (No Response)

Reviewer #4: All comments have been addressed

2. Is the manuscript technically sound, and do the data support the conclusions?

Reviewer #3: Partly

Reviewer #4: Yes

3. Has the statistical analysis been performed appropriately and rigorously? 

Reviewer #3: Yes

Reviewer #4: Yes

4. Have the authors made all data underlying the findings in their manuscript fully available?

Reviewer #3: Yes

Reviewer #4: Yes

5. Is the manuscript presented in an intelligible fashion and written in standard English?

Reviewer #3: Yes

Reviewer #4: No

6. Review Comments to the Author

Reviewer #3: Overall the study addresses a critical issue related to maternal health in Pakistan and offers valuable insights into the utilization of ANC and SBA services. The research design is sound, but would benefit from additional details.

The introductory section exhibits a conflation of themes pertaining to LMICS (Low- and Middle-Income Countries) and Pakistan. In light of this, a more suitable approach would entail a restructuring of the introduction, wherein the discussion initially centers on LMICS and subsequently narrows its focus to Pakistan. Given that the manuscript emphasizes the importance of quality ANC, it would be advantageous for the introduction to be more specifically focused on this aspect of ANC and SBA. However, it lacks a clear statement of the study's objectives and hypotheses and how the research contributed to the already established literature on the topic. This could be beneficial to guide readers through the research focus.

The methods section provides an adequate description of the data source and study variables. However, the section should elaborate on the rationale to support the use IANC. The services described in the constitution of IANC doesnot encompass the broader concept of quality and patient safety. Therefore, it is imperative that the authors explicitly specify the usage of "quality" and "ideal" ANC. It is suggested to provide a sample frame selection framework and number of participants for the two outcomes. It would be beneficial if the univariate and bivariate variables selection criteria for the multivariate model can be stated in the methods sections.

The results section is well documented. However, it is suggested to present descriptive statistics as number and %.

The discussion provides a comprehensive analysis of the study's findings and relates them to existing literature, offering a valuable contribution to the field. However, it needs minor revision to curtail the use of overwhelming statistics. To improve the clarity and readability of the section, the authors should consider organizing the statistics into concise and well-structured sentences. This approach will help the readers better grasp the key findings without getting lost in a sea of numbers. Discussion could be further strengthened by including a critical evaluation of the possible sources of bias. Additionally, the discussion would benefit from a more explicit connection between the study's results and the policy implications proposed.

Reviewer #4: Editing, copywriting and proofing from a Native English language editor is recommended for the paper, for overall language of the manuscript is weak and needs improvement.

7. PLOS authors have the option to publish the peer review history of their article (what does this mean?). If published, this will include your full peer review and any attached files.

Reviewer #3: No

Reviewer #4: No

---

## [Author Response · Author response to Decision Letter 2]

21 Oct 2023

REVIEW COMMENTS TO THE AUTHOR

Reviewer #3: Overall the study addresses a critical issue related to maternal health in Pakistan and offers valuable insights into the utilization of ANC and SBA services. The research design is sound, but would benefit from additional details.

The introductory section exhibits a conflation of themes pertaining to LMICS (Low- and Middle-Income Countries) and Pakistan. In light of this, a more suitable approach would entail a restructuring of the introduction, wherein the discussion initially centers on LMICS and subsequently narrows its focus to Pakistan. Given that the manuscript emphasizes the importance of quality ANC, it would be advantageous for the introduction to be more specifically focused on this aspect of ANC and SBA. However, it lacks a clear statement of the study's objectives and hypotheses and how the research contributed to the already established literature on the topic. This could be beneficial to guide readers through the research focus.

The methods section provides an adequate description of the data source and study variables. However, the section should elaborate on the rationale to support the use IANC. The services described in the constitution of IANC does not encompass the broader concept of quality and patient safety. Therefore, it is imperative that the authors explicitly specify the usage of "quality" and "ideal" ANC. It is suggested to provide a sample frame selection framework and number of participants for the two outcomes. It would be beneficial if the univariate and bivariate variables selection criteria for the multivariate model can be stated in the methods sections.

The results section is well documented. However, it is suggested to present descriptive statistics as number and %.

The discussion provides a comprehensive analysis of the study's findings and relates them to existing literature, offering a valuable contribution to the field. However, it needs minor revision to curtail the use of overwhelming statistics. To improve the clarity and readability of the section, the authors should consider organizing the statistics into concise and well-structured sentences. This approach will help the readers better grasp the key findings without getting lost in a sea of numbers. Discussion could be further strengthened by including a critical evaluation of the possible sources of bias. Additionally, the discussion would benefit from a more explicit connection between the study's results and the policy implications proposed.

Reply:

The introduction and discussion sections have undergone revisions, along with the addition of a few references to support the introduction and explanation of definition for ideal ANC.

Reviewer #4: Editing, copywriting and proofing from a Native English language editor is recommended for the paper, for overall language of the manuscript is weak and needs improvement.

Reply:

The language has been refined. I trust it has been improved.

---

## [Editor Report · Decision Letter 3]

30 Oct 2023

Factors associated with maternal health services utilization in Pakistan: evidence from Pakistan Maternal Mortality Survey, 2019

PONE-D-22-31027R3

Dear Dr. Khan,

We’re pleased to inform you that your manuscript has been judged scientifically suitable for publication and will be formally accepted for publication once it meets all outstanding technical requirements.

Kind regards,

Syed Khurram Azmat, PhD, MPH, MD

Academic Editor

PLOS ONE

---

## [Editor Report · Acceptance letter]

8 Nov 2023

PONE-D-22-31027R3 

Factors associated with maternal health services utilization in Pakistan: evidence from Pakistan Maternal Mortality Survey, 2019 

Dear Dr. Khan:

I'm pleased to inform you that your manuscript has been deemed suitable for publication in PLOS ONE. Congratulations! Your manuscript is now with our production department. 

Kind regards, 

on behalf of

Dr. Syed Khurram Azmat 

Academic Editor

PLOS ONE